# A Variant in *TBCD* Associated with Motoneuronopathy and Corpus Callosum Hypoplasia: A Case Report

**DOI:** 10.3390/ijms241512386

**Published:** 2023-08-03

**Authors:** Maria Caputo, Ilaria Martinelli, Nicola Fini, Giulia Gianferrari, Cecilia Simonini, Rosanna Trovato, Filippo Maria Santorelli, Alessandra Tessa, Jessica Mandrioli, Elisabetta Zucchi

**Affiliations:** 1Department of Biomedical, Metabolic and Neural Sciences, University of Modena and Reggio Emilia, 41125 Modena, Italy; 296988@studenti.unimore.it (M.C.); 177540@studenti.unimore.it (G.G.); 186600@studenti.unimore.it (C.S.); 2Department of Neurosciences, Azienda Ospedaliero-Universitaria Di Modena, Viale Giardini, 1355, 41126 Modena, Italy; martinelli.ilaria88@gmail.com (I.M.); fini.nicola@aou.mo.it (N.F.); elisabetta.zucchi@unimore.it (E.Z.); 3Clinical and Experimental PhD Program, University of Modena and Reggio Emilia, 41125 Modena, Italy; 4Molecular Medicine, IRCCS Fondazione Stella Maris, 56128 Pisa, Italy; rosy.tr@hotmail.it (R.T.); filippo3364@gmail.com (F.M.S.); aletessa@gmail.com (A.T.); 5Neuroscience PhD Program, University of Modena and Reggio Emilia, 41125 Modena, Italy

**Keywords:** tubulin-specific chaperon D, distal motoneuronopathy, tubulinopathy, corpus callosum hypoplasia

## Abstract

Mutations in the tubulin-specific chaperon D (*TBCD*) gene, involved in the assembly and disassembly of the α/β-tubulin heterodimers, have been reported in early-onset progressive neurodevelopment regression, with epilepsy and mental retardation. We describe a rare homozygous variant in *TBCD*, namely c.881G>A/p.Arg294Gln, in a young woman with a phenotype dominated by distal motorneuronopathy and mild mental retardation, with neuroimaging evidence of corpus callosum hypoplasia. The peculiar phenotype is discussed in light of the molecular interpretation, enriching the literature data on tubulinopathies generated from *TBCD* mutations.

## 1. Introduction

The *TBCD* gene is located in the chromosome 17q25.3 and codes for a protein of 1193 amino acids termed tubulin folding cofactor D. The TBCD protein is mainly composed of armadillo/HEAT motifs or multiple α-helices running in antiparallel directions linked by short loops and a TFCD-C C terminal. It contains three HEAT repeats (Figure 1).

The family of tubulins reaches levels of ~5% of total cell protein content and are the constituent of microtubules, essential components of all eukaryotic cells as GTP-binding proteins, involved in a number of functions such as cell division, morphology, polarization, migration, and intracellular transport [1].

TBCD actively participates in the assembly–disassembly of the apical junctional complex and other intercellular and cellular–substratum binding of epithelial cells [2]. In vitro studies showed that the overexpression of cofactors D in cultured cells resulted in the destruction of the tubulin heterodimer and of the microtubules and in ectopic dendrite arborization, suggesting that an optimum level of TBCD is crucial for in vivo neuronal morphogenesis [3]. 

Defective TBCD function impairs soluble α/β-tubulin levels and accelerated microtubule polymerization in patient-derived primary cells [4]. Similarly, the hyperstabilization of microtubules has been shown to result in neurodegeneration, as in the autosomal-dominant spastic paraplegia type 4 (SPG4), which is associated with the degeneration of the corticospinal tracts and heterozygous mutations in SPAST, encoding a protein (spastin) implied in microtubule assembly and dynamics [5]. In vitro investigations examined the protein–protein interactions between the TBCD (wildtype and mutant) and other complex components (a/b-tubulin, TBCE, TBCC, and ARL2), suggesting that the TBCD mutations, which affect the folding of the a-solenoid repeat domains, may impair the proper formation of the microtubule chaperone complexes and consequently of the proper synaptic transmission in the developing brain [1,6,7]. 

Mutations in the genes encoding tubulins and microtubule-associated proteins cause different neurodevelopmental and early-onset neurodegenerative disorders [8,9,10]. They are more commonly described in patients with consanguineous parents, and it accounts for approximately 24% of early-onset neurodegenerative encephalopathies [8]. All the patients described in the literature presented during the first year of life with developmental regression, epilepsy, and microcephaly [8,9,10]. *TBCD* has been associated with atypical Spinal Muscular Atrophy (SMA) only recently, following a single case report [11].

Here, we report a rare homozygotic mutation in the *TBCD* gene in a woman with a phenotype resembling distal motorneuronopathy. We discuss the putative role of this gene in our case in light of the molecular and previous literature data.

## 2. Case Presentation

A 22-year-old woman presented with early fatigue and low exercise tolerance since childhood. She was born from a full-term eutocic birth. On personal history, she reported a slight intellectual disability with an Intelligence Quotient at age 17 measured by WAIS-IV of 61, mainly driven by defective linguistic (63) and comprehension (63) skills, childhood social anxiety disorder with selective mutism and behavioral disorder, and pavor nocturnus until age 8. Her history was negative for epilepsy, and her EEG record was normal. The motor and developmental stages of development were normally achieved. Upon orthopedic assessment, initially the symptoms were ascribed to the feet deformity (flatfeet), leading to bilateral corrective surgery when she was 13. Given the persistence of symptoms after surgery and physical rehabilitation, a neurological examination at 17 years old was performed, showing mild distal weakness, without visible fasciculations, associated with hypotrophy in the intrinsic muscles of the hand and posterior–anterior distal leg muscle lodges (Figure 2A,B). The reflexes were globally normal in the upper limbs, whereas they were absent in lower limbs. The Babinski and Hoffman reflexes were absent. Slight muscle hypotonia at the lower limbs was detected. No sensory or coordination impairment was detected. A mild low-amplitude distal hand tremor was noted.

Neurophysiological study showed sporadic fibrillations potentials and positive sharp waves in the left I dorsal interosseous and right anterior tibial. The motor unit potentials were globally of large amplitude and long duration, with sporadic polyphasic potentials. Decreased motor unit recruitment was revealed, thus revealing a reduced number of motor units. The motor evoked potentials revealed a prolongation in the peripheral motor conduction time.

The cervical Magnetic Resonance Imaging (MRI) was normal, whereas a brain MRI repeated at age 21 revealed a thin corpus callosum (Figure 3), which was previously suspected but could not be definitely determined at age 17, with normal representation of cerebellar morphology and ventricular system.

Slightly elevated creatine kinase (CK) values (217 UI/mL) were reported in the serum. An extensive autoimmune screening was uninformative. In looking for a systemic involvement, the abdomen and thoracic ultrasound and eye examination revealed no abnormalities. The transthoracic heart ultrasound showed a slight left ventricular wall thickening without any dilatation or alterations in segmental kinetics. The ascending aorta and all the valves were normal.

The disease slowly progressed over the years, and the patient mainly complains of fatigue in several daily actions, including climbing the stairs and walking for long distances, without shortness of breath. She did not report any difficulties with the fine movements with her hands. The family history (Figure 4) revealed that her parents were consanguineous, being first cousins, without any intellectual disability or motor signs; their past medical history was overall negative for neurological conditions. Her 25-year-old sister presented a long history of isolated muscle cramps. Her father died from pulmonary artery intimal sarcoma when he was 60 years old. Her mother’s sister was affected by a severe intellectual disability and motor impairment since birth, with development delay, not better investigated. A mother’s first cousin had autism and pes planus.

Informed consent for genetic analysis was obtained from the patient. Genomic DNA was extracted from the peripheral lymphocytes obtained from the patient as standard procedures [12]. The MLPA analysis to search for multiexon rearrangements in the *SPAST* gene and the direct sequencing of the *SMN* (Survival Motor Neuron) gene did not show any deletions or mutations.

A multigene panel including 271 genes explored by Next Generation Sequencing and involved in hereditary spastic paraparesis, motor neuronopathies, and peripheral nerve involvement (see Appendix A) revealed a homozygous point mutation c.881G>A/p.Arg294Gln in *TBCD*. The variant was confirmed by Sanger sequencing, and it was heterozygous in the healthy parents and sister. The mutation was classified as Likely Pathogenic according to the ACMG nomenclature (https://varsome.com/ accessed on November 2021).

## 3. Discussion

Altered TBCD protein levels affect microtubule formation leading to abnormal microtubule trafficking in the human brain, and impairing dendrites’ arborization. Microtubule dynamics are complex tightly regulated processes at the basis of cell viability, architecture, and division. Tubulin alpha and beta assemble in heterodimers following the concerted action of chaperone molecules as TBCA to TBCE; in particular, TBCD-β tubulin and TBCE-α tubulin interact with each other, and together with TBCC they form a supercomplex that releases native tubulin heterodimers upon E-site GTP hydrolysis [13]. Moreover, TBCD activity is modulated by the small GTPase Arl2, which is constitutionally bound to TBCD often forming a trimer of TBCD-β tubulin-Arl2 [14].

Given their fundamental role in regulating cell growth and dendrites formation, tubulinopathies are a heterogenous group of conditions characteristically presenting with cortical malformations and dysmorphic basal ganglia [15,16]. From a review of the literature, different mutations are associated with a broad spectrum of neurological disorders, mainly involving infants with initial normal development followed by neuroregression, epilepsy, and brain atrophy (for complete details, please refer to Table 1). In particular, several compound heterozygous mutations were reported in Chinese and Japanese children presenting with early-onset developmental regression, epilepsy of infancy with migrating focal seizures, and hypotonia. On the brain MRI, cerebral atrophy with secondary microcephaly and brain atrophy with thin corpus callosum were reported [8].

Homozygous mutations were also described in several Israeli, Japanese, Faroese, Indian-Jewish, and Egyptian-Jewish children [1,9], with an overlapping phenotype characterized by muscle weakness, absent visual tracking, and postnatal microcephaly. More than half of the affected individuals had postnatal growth failure, seizure, respiratory failure, developmental regression, optic nerve atrophy, hypotonia, and muscle atrophy [1].

Interestingly, two siblings were diagnosed with atypical SMA at a very early age during childhood, with hypotonia and muscle weakness indicating lower motor unit dysfunction, and progressive complicated central nervous system dysfunctions, axonal-dominant degeneration or motor unit reduction, and pes equinus of both feet was noted. In this case, two different mutations in two different genes were reported, but the homozygous mutation in *TBCD* gene (Arg942Gln) was considered to be the causative variant [11].

The mutation found in our case is likely to be harmful, upon in silico prediction using PolyPhen-2 (Polymorphism Phenotyping at http://genetics.bwh.harvard.edu/pph2/ accessed on November 2021). Upon 3D modeling, we observed that the amino acid change from arginine to glutamine at residue 294 caused a destabilization in the binding site to the complex Arl2/beta tubulin with other residuals in the evolutionary conserved alpha helices structure, altering the physiologic protein function (Figure 5).

Our patient presented a milder phenotype compared to most cases reported in the literature, with clinical findings limited to slowly progressive muscle weakness and slight intellectual retardation along with a neurophysiological pattern of axonal-dominant motor neuron degeneration and motor unit reduction [11]. Interestingly, a similar slight delay in intellectual development was described by Tian et al. [17], although our patient did not manifest symptoms attributable to autism spectrum disorder (ASD) or treatable epilepsy. Curiously, although a mild elevation of CK was recorded in a patient by Chen et al. [18], the clinical picture of neurological involvement was more serious than our case. Additionally, the cerebral MRI revealed thin corpus callosum, similar to the cases of the Chinese and Japanese children described by Zhang and Chen [8,18], confirmed by the data on tubulinopathies association with corpus callosum hypoplasia [15].

In our case, the phenotype was, however, dominated by motor symptoms with neurophysiological evidence of a selective motor neuronopathy at four limbs like SMA. Only recently has the pathogenic link between SMN deficiency and altered microtubule stability been elucidated, pointing to a mitochondrial mislocalization in motor neurons driven by SMN loss [19]. Miyake et al. reported in their case series that patients with a homozygous missense mutation may present a milder phenotype possibly because binding of the altered TBCD protein with β-tubulin was only mildly affected, whereas the patients with most severe phenotype carried truncated or missense variants, which significantly impaired the TBCD binding to other crucial cellular scaffold players [1,6,7]. To this end, Flex et al. confirmed by functional studies in fibroblasts that TBCD mutants may differently impact the complex stabilization with ARL2, TBCE, and β-tubulin, suggesting that the severity of the TBCD loss of function shapes the phenotype along the spectrum of tubulinopathies [4]. In a recent study from South America, two subjects presented with a severe phenotype characterized by developmental encephalopathy and SMA with a novel homozygous missense mutation in TBCD close to a previously described site, which resulted in perturbed TBCD function and microtubule dynamics [20]. To the best of our knowledge, the compound heterozygous mutation c.881G>A in the *TBCD* gene has been only associated with continuous epileptic spasms as a severe form of status epilepticus [21], without mentioning a peripheral nervous system involvement.

Therefore, it could be hypothesized that some *TBCD* missense mutations, such as the one here described, c.881G>A, may only partially disrupt the protein capabilities to bind β tubulin-Arl2 and consequently affect tubulin dynamics more selectively in motor neurons compared to cortical neurons during the developmental stages.

Our case well illustrates how atypical SMA presentations with only mild intellectual impairment should be further inquired for developmental disorders of infancy with central nervous system neuroimaging and extensive genetic panels taking into consideration TBCD and other proteins involved in microtubule dynamics and mitochondrial trafficking. For these selected cases, antisense therapy currently available for SMN1-defective SMA would not be feasible, and other molecules with microtubule-stabilizing functions could be considered. The wide range of neurological effects of *TBCD* gene alterations is complex and growing with the literature data involving mainly neurodevelopment regression and epilepsy. In this context, the molecular basis of these disorders involving the complex tubulin machinery could be better understood through the detection of differential gene variants and their effective clinical impacts.

**Table 1 ijms-24-12386-t001:** The TBCD variants published in the previous literature associated with phenotypes of early-onset neurodegenerative encephalopathy. Abbreviations: ASD: autism spectrum disorders; CC: corpus callosum; CSE: convulsive status epilepticus; EEG: electroencephalogram; F: female; Het: heterozygous; Homo, homozygous; M; male; MRI, magnetic resonance imaging; NCS, nerve conduction studies; NCSE: non convulsive status epilepticus. NR: not reported. WM: white matter. The * indicated the predicted consequence at the protein level of the variant to translation termination codon.

Number of Patients	Sex	Ethnicity	Age at Onset	Familiarity	Zygosity (Het/Homo)	Allele 1 Variant	Allele 2 Variant	Amino Acid Change	Reference	Perinatal History	Neurological Symptoms						Neurological Assessments			Extra-Neurological Manifestations
											Epilepsy	Peripheral Neuropathy	Mental Retardation	Hypotonia	Bulbar Involvement	Others	Imaging/MRI	NCS	EEG	
1	M	Chinese	neonatal period	NR	compound heterozogous	c.881G>A	c.22801C>A	R294Q	Liao, 2020 [21]	NR	Intractable epilepsy, focal seizures	NR	At 20 months: significant atrophy	NR	spared	NR	significant atrophy	NR	Low voltage on EEG, NCSE, CSE	NR
1	M	Chinese	10 months	point mutation in the father, deletion in the mother (affected)	compound heterozygous	230A>G	deletions of exons 28 to 39	H77R	Zhang, 2018 [8]	NR	Intractable epilepsy, generalized seizures	NR	development regression after 5 months of age	yes	spared	microcephaly, hyperreflexia, reduced motor activity, bilateral Babinski reflexes	diffuse cortical atrophy with thinned CC	NR	high-amplitude delta wave background, multi- focal interictal spikes	bilateral hip dislocation at 8 months of age.
1	M	Japanese	at birth	parents were carrier	compound heterozygous	c.1564-12C>G (splicing)	C.2314C>T	R772C	Miyake, 2016 [1]	normal	no	NR	development regression	yes	spared	respiratory failure, muscle atrophy	cortical atrophy	NR	NR	NR
1	F	Japanese	at birth	parents were carrier	compound heterozygous	c.1564-12C>G (splicing)	C.2314C>T	R772C		normal	no	NR	development regression	yes	spared	respiratory failure, muscle atrophy	cortical atrophy	NR	NR	NR
1	F	Japanese	1 month	parents were carrier	compound heterozygous	c.1160T>G	c.2761G>A	M387R		normal	west syndrome	NR	development regression	yes	spared	respiratory failure, muscle atrophy	NR	NR	NR	NR
1	M	Japanese	1 month	parents were carrier	compound heterozygous	c.1160T>G	c.2761G>A	A921T		normal	cataplexy	NR	development regression	yes	spared	respiratory failure, muscle atrophy	NR	NR	NR	NR
1	M	Chinese	5 months	parents were carrier	compound heterozygous	c.2280C>A	c.3365C>T	Y760 *		normal	generalized seizures	NR	development regression	yes	spared	respiratory failure	NR	NR	NR	NR
1	F	Chinese	5 months	parents were carrier	compound heterozygous	c.2280C>A	c.3365C>T	P1122L		normal	generalized seizures	NR	development regression	yes	spared	NR	NR	NR	NR	NR
1	F	Israelian	9 months	parents were carrier	homozigous	c.2810C>G		P937R		normal	generalized seizures	NR	development regression	no	spared	NR	NR	NR	NR	NR
1	F	Israelian	9 months	parents were carrier	homozigous	c.2810C>G		P937R		normal	generalized seizures	NR	development regression	no	spared	NR	NR	NR	NR	NR
1	F	Japanese	5 months	parents were carrier	homozigous	c.2825G4A		R942Q	Ikeda, 2016 [11]	normal	partial seizure	NR	development regression	yes	spared	NR	cortical atrophy	NR	NR	NR
8	M	Faroese	6 months	parents of 2 patients were carrier	homozigous	3099C>G		N1033K	Grønborg, 2018 [10]	normal	generalized treatment resistant epilepsy	NR	development regression	yes	spared	respiratory failure, spasticity,	cortical and global cerebral atrophy	NR	NR	Bilateral hip luxation
1	M	Indian-Jewish	20 months	parents were heterozigous carrier	homozigous	c.1423G*>*A		A475T	Pode-Shakked, 2016 [9]	normal	generalized seizures	NR	development regression	yes	spared	microcephaly and right-sided plagiocephaly,	dilatated ventricles and subarachnoid spaces with diffuse thinning of the WM and CC, mild secondary hypomyelination	NR	disorganized high amplitude delta wave background and multi- focal polyspike discharges at moderate rate.	low anterior hairline, large ears, pectus excavatum, right hand single transverse palmar crease, lateral deviation of the first toes
1	F	Egyptian-Jewish	24 months	consanguineous, carrier	homozigous	c.2810C*>*G		P937R		normal	generalized seizures	NR	development regression	no	spared	NR	mild cortical atrophy, moderately thin corpus callosu	normal	high-amplitude delta wave background, multi- focal interictal spikes	
1	F	Egyptian-Jewish	24 months	consanguineous, carrier	homozigous	c.2810C*>*G		P937R		normal	generalized treatment resistant epilepsy	NR	development regression	no	spared	NR	cortical atrophy and moderately thin CC	NR	NR	NR
1	M	German/Sicilian/Cajun-Hungarian/Irish	6 months	parents were heterozigous carrier	compound heterozigous	c.1757C*>*T	c.3192-2A>G	A586V		normal	generalized treatment resistant epilepsy	severe motor axonal neuropathy	development regression	yes	spared	NR	cortical atrophy and moderately thin CC	severe motor axonal neuropathy	NR	NR
1	M	German/Sicilian/Cajun-Hungarian/Irish	6 months	parents were heterozigous carrier	compound heterozigous	c.1757C*>*T	c.3192-2A>G	A586V		normal	generalized treatment resistant epilepsy	NR	development regression	no	spared	NR	cortical atrophy and moderately thin CC	NR	NR	several tendon lengthening orthopedic surgeries
1	F	Chinese	12 months	NR	compound heterozigous	c.3365C>T	c.1739G>A	P1122L, R580Q	Tian, 2019 [17]	NR	generalized tonic-clonic seizures	NR	slight delay of intellectual development	no	NR	dystonia	myelination delay reflected by abnormal signal in the occipital WM	NR	Interictal EEG: large number of spike waves	NR
1	F	Chinese	6 months	NR	compound heterozigous	c.230A>G	c.907C>T	H77R, R303 *		NR	generalized tonic-clonic seizures	NR	nearly normal intellectual development	no	NR	NR	myelination delay reflected by abnormal signal in the occipital WM	NR	Interictal EEG: low amplitudespike waves in midline	NR
1	M	Chinese	-	NR	compound heterozigous	c.2953C>T	c.3550C>T	R979C, Q1184 *		NR	generalized tonic-clonic seizures	absent	ASD	no	NR	NR	normal	NR	NR	NR
1	F	Chinese	12 months	parents were heterozigous carrier	compound heterozigous	c.1340C>T	c.817+2T>C	A447V	Chen, 2021 [18]	normal	generalized tonic-clonic seizures	NR	early-onset neurodegeneration, failure to thrive	yes	failure to thrive	respiratory failure	thinning of the CC, diffuse cerebral atrophy involving both gray and WM, dilatated ventricles	NR	NR	severe scoliosis, thrombocytopenia, presence of accessory spleen
1	F	Chinese	18 months	parents were heterozigous carrier	compound heterozigous	c.1340C>T	c.817+2T>C	A447V		normal	focal to generalized tonic-clonic seizures	NR	early-onset neurodegeneration	yes	spared	respiratory failure	hypoplasia of CC, prominent enlargement of cerebral cortical sulci and ventricles	NR	slow wave activities	mild elevation of aspartate aminotransferase and CK (335 IU/L).

## 4. Conclusions

Our findings extend the phenotypic traits of tubulinopathy generated from *TBCD* mutations, by the identification of a rare homozygous *TBCD* variant associated with a predominant distal motoneuronopathy, slight mental retardation, and corpus callosum atrophy, a clinical picture distinct from the literature reports. In this setting, computational modeling methods and accurate genotype–phenotype correlations may better clarify the impact of the secondary and tertiary structure alteration following amino acid substitution, to better understand the mechanism of neurodegeneration related to tubulinopathies, leading to an early diagnosis and appropriate genetic counseling.

## Figures and Tables

**Figure 1 ijms-24-12386-f001:**
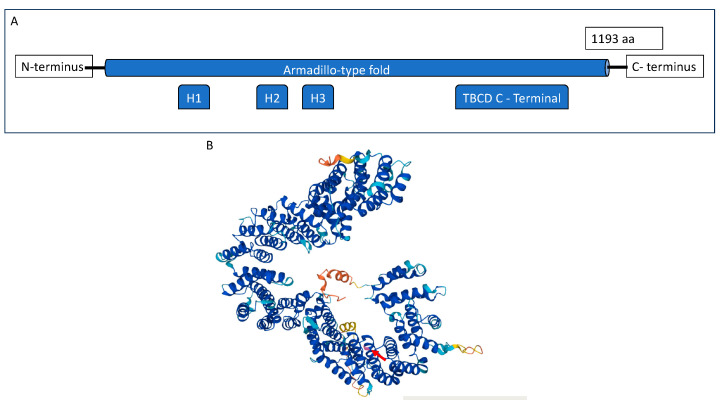
(**A**) Schematic representation of the TBCD protein. (**B**) Three-dimensional structure of the TBCD protein. The protein is mainly composed of α-helices in antiparallel configuration, typical of armadillo/HEAT motifs. Arg294 (pointed by the red arrow) is represented in light purple, and it is located in an alpha helix domain.

**Figure 2 ijms-24-12386-f002:**
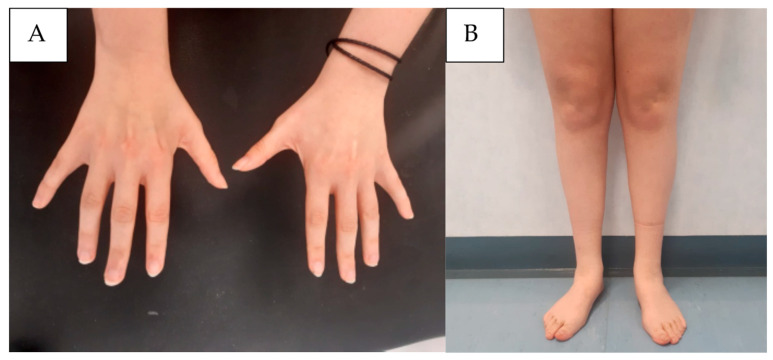
Proband’s clinical picture, showing atrophy at the level of the intrinsic hand muscles (**A**) and an inverted “champagne-bottle” appearance to the lower extremities (**B**).

**Figure 3 ijms-24-12386-f003:**
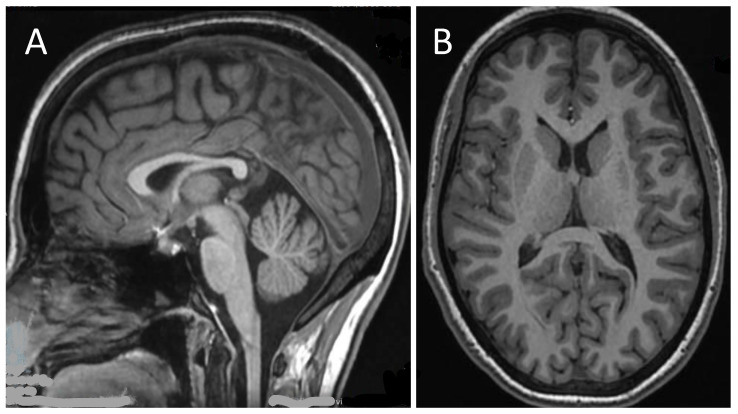
Brain magnetic resonance imaging (MRI) of the patient at 17 years of age: sagittal T1-weighted image revealed thinned corpus callosum (**A**); axial T1-weighted image shows normal morphology of the ventricular system (**B**).

**Figure 4 ijms-24-12386-f004:**
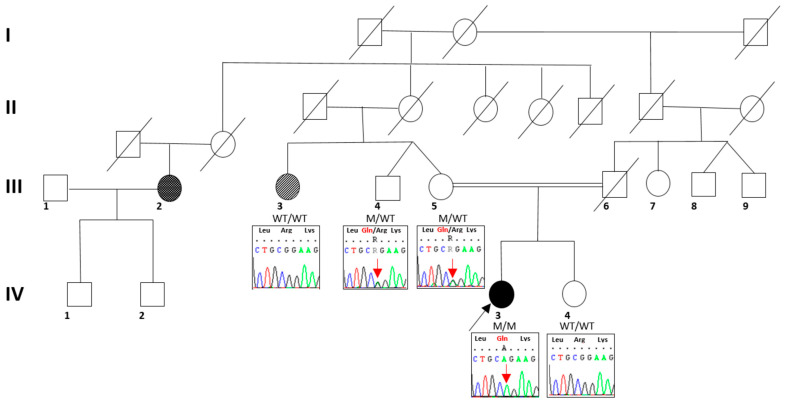
Proband’s pedigree (indicated by an arrow, IV—3). The filled symbols indicate individuals affected by neurological disorders. The grey color indicates the proband’s aunt, presenting with intellectual disability since birth and motor impairment since 8 months old; the black color indicates the proband’s second-degree cousin presenting with autism and pes planus; no segregation analysis was available for him.

**Figure 5 ijms-24-12386-f005:**
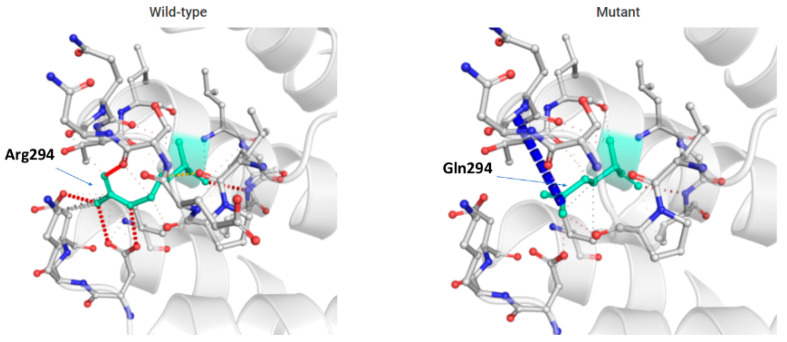
In silico prediction of the functional consequence of c.881G>A in the *TBCD* gene, with substitution of arginine with glutamine at position 294. Wildtype and mutant residues are colored in light green and are also represented as sticks alongside with the surrounding residues, which are involved in any type of interactions.

## Data Availability

Further data concerning the subject clinical presentation are available upon reasonable request to the corresponding author.

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
