# Peer review of "A Variant in TBCD Associated with Motoneuronopathy and Corpus Callosum Hypoplasia: A Case Report"

_ijms, 2023, doi:10.3390/ijms241512386_

Round 1
Reviewer 1 Report
Ms. ID #2508624
The AAs described an apparently novel homozygous variant linked to TBCD gene and related to distal motor neuropathy, mild ID deficit, and corpus callosum hypoplasia in a young female patient. The AAs have made significative efforts in the reviewing of the literature even so it should be updated and discussed. Introduction seems to be quite appropriate even so the AA would emphasize the role of TBCD gene in synaptic transmission. It is known the prevalence of the disease ? Is a genotype-phenotype relationship known ?
Major criticisms are related to article title, case presentation, literature data reviewing (Table 1 and 2) presentation, Discussion section and literature.
Article title. The AA described a new variant (c.881G>A) associated to motoneuronopathy and corpus callosum hypoplasia in a young female patient. The variant was previously identified by Liao et al. 2020 and reported in ClinVar database (Accession VCV000449795.3). I do not understand the use of the adjective “New”.
Is the aim of the manuscript the description of new genetic variant or new clinical feature (motoneuronopathy) in the known variant of TBCD gene? If so, in my opinion, the title should be modified. The AAs would clarify this point.
Case description is rather confused whereas the family history is sufficient clear. Did the parents (or mother) of the patient show clinical features possibly related to borderline ID/mild ID, learning difficulties and/or motor signs?
The AA reported ..”All the patients described …during the first year of life with developmental regression, epilepsy, and microcephaly..:” but these informations did not reported in the description. As ASD is identified as a novel clinical phenotype of TBCD-related disorders (Tian D et al. 2019), did the AA assess this aspect? Furthermore, as the literature reported the presence of microcephaly, epilepsy, the AAs should provide some information in order to provide a clear picture of the female patient and a clearer timeline of events. In particular:
1) possible pregnancy problems and perinatal/neonatal data
2) developmental milestones besides motor milestone
3) anthropometric data and head circumference
4) presence of fine motor ability (i.e., purposeful hand use), muscle hypotonia
5) presence of a possible deficit in social interactions, ASD-like behavior, learning problems and language skill
6) presence of epilepsy or epilepsy history
7) presence of respiratory symptoms
8) EEG record
9) Figure 4: lacking information regarding the type of neurological disorders reported for the subjects identified with filled symbols
Furthermore, the AA reported “…Slightly elevated CK values were reported in serum.. Transthoracic heart ultra-94 sound showed a slight parietal thickness..”. The AA should describe these findings.
From Table 1 and 2 resulted that subjects harboring homozygous variants for TBCD gene showed epilepsy and developmental regression. How do you explain the absence of these clinical features in your patient?
I observed that other homozygous gene variants already reported were localized on the TBCD gene starting from nucleotide #1423. Could this element be important for the pathophysiological alteration of protein and then implication on the phenotypic picture ?
Table 1 and 2. Presentation should be improved. To a clearer readership and data interpretation, I would suggest re-organization of the items (i.e., n. patients, sex, ethnicity, age at onset, familiarity, zygosity, allele1 variant, allele 2 variant, aa change, and reference) and combine the 2 tables. In order to address this point, I suggest to abbreviate “not reported” in NR. Furthermore, the Term MRI should be placed in the column title “Imaging/MRI”. The new Table should be implemented with the new 2 articles.
Discussion.
Lines 117 to 123 description of a possible pathogenicity should be better placed in the Discussion section.
Lines 203-209 “..In conclusion,…” this sentence should be placed in the Conclusion section. All the conclusion section should be reduced in 1-2 sentence/s. The findings of the two new studies should be discussed.For this reason, the discussion should be re-organized. I did not understand the use of semicolon (lines 182-191)
Literature should be updated with two new articles and the findings included in the Ms.
Tian D, Rizwan K, Liu Y, Kang L, Yang Y, Mao X, Shu L. Biallelic pathogenic variants in TBCD-related neurodevelopment disease with mild clinical features. Neurol Sci. 2019 Nov;40(11):2325-2331
Chen CL, Lee CN, Chien YH, Hwu WL, Chang TM, Lee NC. Novel Compound Heterozygous Variants in TBCD Gene Associated with Infantile Neurodegenerative Encephalopathy. Children (Basel). 2021 Dec 5;8(12):1140.
Minor criticisms:
-Many grammar and styling errors in the Discussion.
-Some terms have been abbreviated after the first citation in the text (i.e., SMA) whereas for other terms (i.e., PSW and MUP) the full name is missing.
-Faroese ethnicity in the place of Faraose ethnicity
-Neonatal period in the place of neonatal age
-Age at onset in the place of age of onset
Moderate editing of English language is required due to presence of many grammar and styling errors.
Author Response
Reviewer 1
Ms. ID #2508624
The AAs described an apparently novel homozygous variant linked to TBCD gene and related to distal motor neuropathy, mild ID deficit, and corpus callosum hypoplasia in a young female patient. The AAs have made significative efforts in the reviewing of the literature even so it should be updated and discussed. Introduction seems to be quite appropriate even so the AA would emphasize the role of TBCD gene in synaptic transmission.
We thank the reviewer for having highlighted this aspect; we added a sentence in the Introduction in lines 55-56 as follows:
“[…] suggesting that TBCD mutations which affect the folding of the a-solenoid repeat do-mains, may impair the proper formation of microtubule chaperone complexes and consequently of proper synaptic transmission in the developing brain [1,6,7].”
It is known the prevalence of the disease ? Is a genotype-phenotype relationship known ?
We thank you the reviewer for this comment and we value their suggestion as to improve the bibliography behind our work.
In the literature we couldn’t find any work assessing the exact prevalence of biallelic TBCD mutations in determining infantile encephalopathy and a clear-cut relationship between genotype and phenotypic manifestations is currently unknown. In Orphanet the authors reported an estimated prevalence of <1: 1,000,000 of “early-onset diffuse brain atrophy-microcephaly-muscle weakness-optic atrophy syndrome” caused by autosomal recessive mutations in TBCD (only 39 cases reported, https://www.orpha.net/consor/cgi-bin/OC_Exp.php?lng=EN&Expert=496641).
Major criticisms are related to article title, case presentation, literature data reviewing (Table 1 and 2) presentation, Discussion section and literature.
Article title. The AA described a new variant (c.881G>A) associated to motoneuronopathy and corpus callosum hypoplasia in a young female patient. The variant was previously identified by Liao et al. 2020 and reported in ClinVar database (Accession VCV000449795.3). I do not understand the use of the adjective “New”.
Is the aim of the manuscript the description of new genetic variant or new clinical feature (motoneuronopathy) in the known variant of TBCD gene? If so, in my opinion, the title should be modified. The AAs would clarify this point.
Thank you for pointing out this possible mistake; as the referee inferred, we used the term “new” to describe the novel association between c.881G>A variant and motoneuronopathy, which in Liao et al. was described in patients with intractable seizures. Clearly, we understand this may generate misunderstandings and accordingly, we deleted the term “new” from the title; in the text we’ve substituted “novel” in favor of “rare”.
Case description is rather confused whereas the family history is sufficient clear. Did the parents (or mother) of the patient show clinical features possibly related to borderline ID/mild ID, learning difficulties and/or motor signs?
We apologize we did not fully report the family history. Her parents were negative for any intellectual disability or motor impairment; we added this in the text as follows in lines 113-115:
“Family history (Figure 4) revealed that her parents were consanguineous, being first cousins, without any intellectual disability or motor signs; their past medical history was overall negative for neurological conditions.”
The AA reported ..”All the patients described …during the first year of life with developmental regression, epilepsy, and microcephaly..:” but these informations did not reported in the description. As ASD is identified as a novel clinical phenotype of TBCD-related disorders (Tian D et al. 2019), did the AA assess this aspect?
The patient was followed by the local Neuropsychiatric Unit since childhood, when she manifested abnormal behaviour upon stressful situations, as selective mutism. From her neuropsychiatric medical records no ASD was reported, though we haven’t the access to the tests performed at that time.
She then performed regular IQ test which confirmed the slight intellectual disability we now added in the text.
Furthermore, as the literature reported the presence of microcephaly, epilepsy, the AAs should provide some information in order to provide a clear picture of the female patient and a clearer timeline of events. In particular:
1) possible pregnancy problems and perinatal/neonatal data
2) developmental milestones besides motor milestone
3) anthropometric data and head circumference
4) presence of fine motor ability (i.e., purposeful hand use), muscle hypotonia
5) presence of a possible deficit in social interactions, ASD-like behavior, learning problems and language skill
6) presence of epilepsy or epilepsy history
7) presence of respiratory symptoms
8) EEG record
9) Figure 4: lacking information regarding the type of neurological disorders reported for the subjects identified with filled symbols
Thank you for pointing out all these important phenotypic data. We tried our best to add this information in the text, here is a point-by-point answer:
- No pregnancy problem was reported. We had no access to neonatal clinical data as she came to our attention when 17 years old and we could not track these data
- Developmental milestones were reported as regularly achieved by the local Neuropsychiatric Unit who followed the patient since childhood;
- Anthropometric data were not trackable by the time the patient came to our attention
- Fine motor abilities were normally achieved according to the neuropsychiatric evaluations
- Concerning possible deficits in social interactions, as mentioned in the previous questions, aberrant social behaviours were reported as transitional during stressful situations and a diagnosis of ASD were never established by our Neuropsychiatric Unit, though we could not track any specific tests she performed at that time
- No epilepsy history was reported
- No respiratory symptoms was reported and her functional respiratory test by which she is regularly evaluated are in range (last FVC in 2022: 112%)
- Her EEG recording was normal
- Thank you for stressing the incomplete data on the figure; we included the following information in the figure legend (lines 122-125): “Grey color indicates proband’s aunt, presenting with intellectual disability since birth and motor impairment since 8-months old; black color indicates the proband’s second-degree cousin presenting with autism and pes planus; no segregation analysis was available for him”.
Overall, we included all these information in the following paragraph (lines 70-77):
“She was born from a full-term eutocic birth. On personal history she reported a slight intellectual disability with an Intelligence Quotient at age 17 measured by WAIS-IV of 61,mainly driven by linguistic (63) and comprehension (63) defective skills, childhood social anxiety disorder with selective mutism and behavioral disorder, and pavor nocturnus until 8 years old. Her history was negative for epilepsy and her EEG record was normal. Motor and developmental stages of development were normally achieved.”
Furthermore, the AA reported “…Slightly elevated CK values were reported in serum.. Transthoracic heart ultra-94 sound showed a slight parietal thickness..”. The AA should describe these findings.
Apologies for having skipped this important data. We added the following in lines 107-109:
“Transthoracic heart ultrasound showed a slight left ventricular wall thickening without any dilatation or alterations in segmental kinetics. Ascending aorta and all the valves were normal.”
From Table 1 and 2 resulted that subjects harboring homozygous variants for TBCD gene showed epilepsy and developmental regression. How do you explain the absence of these clinical features in your patient?
Thank you for noting this point. We think this represents the true novelty of our case report and we value this observation. In general, independently of the zygosity the phenotypic landscape of biallelic TBCD mutations is rather grim with early onset developmental regression and intractable epilepsy. From the review in the literature we could note other patients presented with biallelic TBCD mutations and a less severe phenotype with less pronounced MRI abnormalities (as in the two siblings CMH444 CMH445 reported by Flex et al), however a history of focal epilepsy was noted. We think the absence of focal or intractable epilepsy is correlated with the mild alterations observed by neuroimaging, with only slight callosal hypoplasia and exclusion of cortical malformations which may favor epilepsy onset. Overall, we may speculate homozygous c.881G>A variant may impact less on microtubule dynamics during developmental stages and consequently cortical and white matter structural organization and synaptic transmission. Patients presenting with homozygous mutations in TBCD represent an exceptional opportunity to study in vivo the functional consequences of each single point mutation in this gene; however, we hope in the future further models will be employed to investigate the structural and functional consequences of this TBCD variant at cellular and phenotypic level.
I observed that other homozygous gene variants already reported were localized on the TBCD gene starting from nucleotide #1423. Could this element be important for the pathophysiological alteration of protein and then implication on the phenotypic picture ?
As above, this is a really interesting point of the whole discussion about the pathogenicity of this variant.
We are not sure we understood in full this comment. Did the expert reviewer put forward the hypothesis that homozygous variants in the TBCD gene occurring around and after nucleotide 1423 in the coding sequence (that is, c.1423G>A/p. Ala475Thr and the other downstream) are relevant in genotype-phenotype correlations? To answer this question, we should have had available a more detailed functional mapping and assessment (even in heterologous systems) of the different missense, data not available in the literature and far beyond the scope of this manuscript. By considering clinical data alone, we observed that there is no clearcut genotype-phenotype correlation in TBCD-related disorders. The p. Ala475Thr leads to a severe developmental disorder and encephalopathy similar to the more distant p. Pro937Arg variant and both show features seen also in children with compound heterozygous variants occurring before or after the c.1423G, though there is a trend to a relatively later age at onset (after the first 18 months). This, however, should be confirmed in larger studies.
Table 1 and 2. Presentation should be improved. To a clearer readership and data interpretation, I would suggest re-organization of the items (i.e., n. patients, sex, ethnicity, age at onset, familiarity, zygosity, allele1 variant, allele 2 variant, aa change, and reference) and combine the 2 tables. In order to address this point, I suggest to abbreviate “not reported” in NR. Furthermore, the Term MRI should be placed in the column title “Imaging/MRI”. The new Table should be implemented with the new 2 articles.
We thank the reviewer for these suggestions. We’ve modified the table accordingly to them, combining the two tables. We hope we can add this table in the main manuscript, but unfortunately we could not fit the file into the Word template provided.
Discussion.
Lines 117 to 123 description of a possible pathogenicity should be better placed in the Discussion section.
Thank you for suggesting this change in placement. We’ve added it before discussing or case in light of literature.
Lines 203-209 “..In conclusion,…” this sentence should be placed in the Conclusion section. All the conclusion section should be reduced in 1-2 sentence/s. The findings of the two new studies should be discussed. For this reason, the discussion should be re-organized. I did not understand the use of semicolon (lines 182-191)
Thank you for proposing this organization of the text. We’ve modified the use of semicolons in favor of comma. Moreover, we’ve simplified the conclusion section. We’ve added the two articles in the discussion as follows: “Interestingly, a similar slight delay of intellectual development, have been described by Tian et al [17], although in our patient did not manifest symptoms attributable to autism spectrum disorder (ASD) or treatable epilepsy. Curiously, although a mild elevation of CK has been recorded also in a patient by Chen et al [18], the clinical picture of neurological involvement was more serious than our case.”
Literature should be updated with two new articles and the findings included in the Ms.
Tian D, Rizwan K, Liu Y, Kang L, Yang Y, Mao X, Shu L. Biallelic pathogenic variants in TBCD-related neurodevelopment disease with mild clinical features. Neurol Sci. 2019 Nov;40(11):2325-2331
Chen CL, Lee CN, Chien YH, Hwu WL, Chang TM, Lee NC. Novel Compound Heterozygous Variants in TBCD Gene Associated with Infantile Neurodegenerative Encephalopathy. Children (Basel). 2021 Dec 5;8(12):1140.
Minor criticisms:
-Many grammar and styling errors in the Discussion.
-Some terms have been abbreviated after the first citation in the text (i.e., SMA) whereas for other terms (i.e., PSW and MUP) the full name is missing.
-Faroese ethnicity in the place of Faraose ethnicity
-Neonatal period in the place of neonatal age
-Age at onset in the place of age of onset
Thank you for pointing out all these typos; we tried our best to revise the styling which we hope have now met the standard of quality of the paper.

Reviewer 2 Report
The current case report is included describing of Tbcd associated with motoneuronopathy and corpus callosum hypoplasia. Authors first described TBCD gene and role of it in neuronal morphogenesis, then they reported a novel homozygotic mutation in the TBCD gene in a girl with a phenotype resembling distal motorneuropathy. It is a very interesting paper but there are some important issues that are needed to be considered:
1. In title authors mentioned case report and literature review, which is confusing, it is better author delete literature review and just keep case report, otherwise it needs include much more information for the literature review.
2. Introduction is only included information about Tbcd gene, however, it is important to add some information about reported cases of Tbcd gene mutation. I suggest to move some information from discussion to the introduction.
3. In some parts of paper, author mentioned girl and some parts woman, it is better to use woman for all parts of the report, since the age of case is 22 years old.
4. If the case had intellectual disability, how authors made sure that the provided information by her was accurate?
5. Authors reported MRI from 17 years old with thinned corpus callosum, do they have more recent MRI close to 22 years old? It is important to have MRI from the time of medical examination.
Author Response
Reviewer 2
The current case report is included describing of Tbcd associated with motoneuronopathy and corpus callosum hypoplasia. Authors first described TBCD gene and role of it in neuronal morphogenesis, then they reported a novel homozygotic mutation in the TBCD gene in a girl with a phenotype resembling distal motorneuropathy. It is a very interesting paper but there are some important issues that are needed to be considered:
- In title authors mentioned case report and literature review, which is confusing, it is better author delete literature review and just keep case report, otherwise it needs include much more information for the literature review.
Thank you for this suggestion. In line with your suggestions and those of the first referee, we modified the title.
- Introduction is only included information about Tbcd gene, however, it is important to add some information about reported cases of Tbcd gene mutation. I suggest to move some information from discussion to the introduction.
Dear reviewer, thank you for the advice. For the sake of the discussion about the phenotypic unicity of our case associated with TBCD biallelic mutations, we added more information in the Discussion section. In fact, we think the peculiarity of our case is represented by the mild phenotype with very slight encephalopathic traits and the prominent motoneuronopathy observed by neurophysiological testing. Accordingly, to better address the reader towards this peculiarity, we added a description of the few cases of atypical SMA reported in the literature in line (62-64):
“TBCD was associated with atypical Spinal Muscular Atrophy only in most recent years, following a single case report [11].”
- In some parts of paper, author mentioned girl and some parts woman, it is better to use woman for all parts of the report, since the age of case is 22 years old.
Thank you for this suggestion. Accordingly, we’ve used the term “woman” instead of “girl”.
- If the case had intellectual disability, how authors made sure that the provided information by her was accurate?
Thank you for the suggestion. We reported what we could track from the neuropsychiatric documentation starting from her late childhood. We better specified her intellectual disability in lines 70-73:
“On personal history she reported a slight intellectual disability with an Intelligence Quotient at age 17 measured by WAIS-IV of 61,mainly driven by linguistic (63) and comprehension (63) defective skills,…”
- Authors reported MRI from 17 years old with thinned corpus callosum, do they have more recent MRI close to 22 years old? It is important to have MRI from the time of medical examination.
We apologize for the lack of clarity. The proband performed a first MRI at age 17, which according to our NeuroImaging Unit was uninformative since she was not 18 and myelination stages could be incomplete by the time. Next, we repeated brain MRI at age 21, as now specified in the text, which revealed still hypoplasic corpus callosum. We added in the text the following (line 96-99):
“(…)whereas a brain MRI repeated at age 21 revealed a thin corpus callosum (Figure 3), which was previously suspected but could not be definitely determined at age 17, with normal representation of cerebellar morphology and ventricular system.”
